# Plant Disease Detection and Classification Method Based on the Optimized Lightweight YOLOv5 Model

**Haiqing Wang, Shuqi Shang \*, Dongwei Wang, Xiaoning He, Kai Feng and Hao Zhu**

College of Mechanical and Electrical Engineering, Qingdao Agricultural University, Qingdao 266109, China; 20202204018@stu.qau.edu.cn (H.W.); 200701031@qau.edu.cn (D.W.); 201502004@qau.edu.cn (X.H.); 20192204162@stu.qau.edu.cn (K.F.); 20202204029@stu.qau.edu.cn (H.Z.)
\* Correspondence: sqshang@qau.edu.cn

**Abstract:** Traditional plant disease diagnosis methods are mostly based on expert diagnosis, which easily leads to the backwardness of crop disease control and field management. In this paper, to improve the speed and accuracy of disease classification, a plant disease detection and classification method based on the optimized lightweight YOLOv5 model is proposed. We propose an IASM mechanism to improve the accuracy and efficiency of the model, to achieve model weight reduction through Ghostnet and WBF structure, and to combine BiFPN and fast normalization fusion for weighted feature fusion to speed up the learning efficiency of each feature layer. To verify the effect of the optimized model, we conducted a performance comparison test and ablation test between the optimized model and other mainstream models. The results show that the operation time and accuracy of the optimized model are 11.8% and 3.98% higher than the original model, respectively, while F1 score reaches 92.65%, which highlight statistical metrics better than the current mainstream models. Moreover, the classification accuracy rate on the self-made dataset reaches 92.57%, indicating the effectiveness of the plant disease classification model proposed in this paper, and the transfer learning ability of the model can be used to expand the application scope in the future.

**Keywords:** disease classification; YOLOv5 model; attention mechanism; transfer learning

## 1. Introduction

Plant diseases are one of the direct threats to food security. The annual global agricultural losses caused by diseases and insect pests are as high as USD 250 billion. In 2021, the area affected by major diseases and insect pests in China had reached 400 million hectares. Therefore, it is crucial for agricultural production to detect crop diseases in a timely manner [1]. One of the more effective solutions is to predict and classify crop diseases by collecting images of crop growth and building a detection model [2], which can give timely and effective early warning and intervention to agricultural production, improve the efficiency of disease control, and reduce agricultural losses caused by diseases.

The traditional disease diagnosis method mainly judges the health status or disease type of crops. It guides agricultural production by manually observing the color, size, and disease spot shape of crop leaves, which has problems, such as high professional requirements, long diagnosis time, and low work efficiency [3]. Based on this, scholars are committed to identifying the characteristics of crop leaves and other characteristics through network models to achieve rapid detection and classification of diseases. At the same time, they also continue to expand the types of datasets to improve the scope of application of the method, which plays a vital reference and guiding role in the quality and output. The early method of extracting plant leaf features was to manually extract feature maps. Researchers [4] extracted feature information from the segmented images and discriminated against abnormal situations. For example, the Otsu threshold method is combined with the K-means clustering method [5] to segment green pixels, and the disease types are identified by texture features. According to the characteristics of rice brown

spot, leaf blast, and bacterial blight, Mohan et al. [6] used the K-nearest neighbors (KNN) algorithm to identify the characteristics of field crops and proposed a method to identify plant diseases in rice fields. To identify and evaluate the incidence of black spot disease, Zhou et al. [7] proposed a multi-feature-based machine learning method, introducing Gabor and Canny operators to extract texture features to determine the incidence of black spot disease in random samples. The above disease detection methods usually require professionals to extract professional features (such as leaf digital features, texture features, and SIFT features) and diagnose diseases. The modeling process is complex, the application is difficult, and the method's applicability needs to be further improved. Therefore, the traditional methods cannot perform the disease detection and classification of crops well.

Object detection technology has become one of the current research hotspots based on the above problem [8–10]. The technology classifies and defines the category of the object by locating and predicting the position of the object to complete the recognition, positioning, and classification of objects in images or videos. Object detection algorithms are mainly divided into two-types (ResNet, LeNet-5 [11], AlexNet [12], GoogLeNet [13], and Faster-RCNN [14]) and single-stage (such as SSD, VGG [15], You Only Look Once (YOLO) [16], etc.).

Although the candidate region-based detection method has relatively high accuracy, there are limitations due to the method's high computational effort and low real-time performance. Therefore, scholars have proposed a single-stage target detection algorithm to achieve fast and efficient detection. For example, Liu et al. [17] proposed the Single Shot MultiBox Detector (SSD) model, which improved the detection accuracy by using the features of different feature maps to identify different positions of the picture. However, using multiple layers of features for recognition makes SSD ignore the correlation between image datasets. In November 2015, Redmon et al. [16] proposed the YOLO model. The detection speed, accuracy, and recognition category of the model are improved by using DarkNet for feature detection after the convolutional layer and using a lower billion floating-point number operation (BFLOP). Based on the mosaic data enhancement method of the YOLOv4 model, in June 2020, Ultralytics et al. [18] added some new ideas for improvement and proposed the YOLOv5 model, which can flexibly control the size of the model by adding the Hardwish activation function. While improving the running speed and accuracy of the model, the memory footprint of the model is greatly reduced, which meets the needs of this study. Based on the above comparative analysis of the performance characteristics of different types of detection algorithms, this study proposes an optimized lightweight YOLOv5 model suitable for plant disease identification tasks by improving and optimizing the YOLOv5 model.

## 2. Related Work

With the expansion of the number of various public datasets and the development of image processing and object detection technologies, the research on leaf disease feature location and classification based on field crop images has developed rapidly [19–23]. The first step of the detection process is the localization of the disease, which is mainly based on the feature information extracted by the model to locate the disease spots and judge the degree of infection. For example, Dyrmann et al. [24] used a fully connected CNN network for weed detection, which could locate and segment weeds from complex environments. By labeling a dataset containing 18,222 images, DeChant et al. [25] built a convolutional neural network model to identify maize leaf blight with 96.7% accuracy. The second step of object detection is the disease classification, which is mainly by matching the given input feature vector with one of all the classes learned during training to obtain the class to which the object belongs. Wang et al. [26] proposed a data-balanced, Faster R-CNN-based identification method for winter jujubes with different maturity in the natural environments according to different shooting angles. Picon et al. [27] increased the diversity of the dataset by randomly adding the background of the ILSVRC15 database to the leaf images and used the ResNet-50 residual network for the automatic identification of wheat diseases. To

achieve rapid detection of tea buds, Xu et al. [28] combined the rapid detection capability of YOLOv3 to propose a detection and classification method with a two-level fusion network with a variable universe, and the recognition accuracy reached 95.71%. Aiming at the relatively small area of plant disease lesions, Liu et al. [29] combined the characteristics of GoogLeNet and AlexNet networks to construct a cascade network with smaller convolution kernels. The related models are introduced in Table 1. However, due to the problems of many target areas and similar target types in the process of plant disease detection and the introduction of a self-made flower disease dataset in this study, the network identification accuracy and speed are required to be high. Therefore, it is necessary to optimize and improve the existing network model to meet the detection needs of this research.

**Table 1.** Comparison of different network models (the generation time, network level, and structural features of different models).

| Model | Generation Time | Network-Level | Structural Features | Advantages and Limitations |
|---|---|---|---|---|
| LeNet | 1998 | 5 | Local connection and weight sharing | Poor processing power for larger image datasets |
| AlexNet | 2012 | 8 | Dropout was proposed to prevent overfitting | High requirements for GPU |
| GoogLeNet | 2014 | 22 | The Inception block was adopted as the base convolutional block | Too much computation |
| VGGNet | 2014 | 19 | Max pooling was used between layers | High memory usage |
| Faster R-CNN | 2016 | - | RPN candidate box generation algorithm | Low detection speed |
| YOLOv1 | 2015 | 26 | DarkNet was adopted for feature detection | Low positioning accuracy |
| YOLOv5 | 2020 | 640 | Hardwish activation function was added | Insufficient attention to small goals and key areas |

Currently, object detection technology has been used in strawberries [30–32], grapes [33], apple fruits [34,35], flowers [36], maize [25], and rice [37,38], and has achieved relatively good application results. There is scientific evidence about the accuracy of the proposed architecture described in the studies by Li et al. [39], in which YOLO-JD has achieved the best detection accuracy, with an average mAP of 96.63%. A similar finding was reported by Islam et al. [40], who obtained an accuracy of 99.43%, but use transfer learning to train the CNN, proposed by the authors, using VGG16. Moreover, the research by Lee et al. [41] obtained an accuracy of 99%; in this study, an adjustment was made to the images, removing the background and leaving only the potato leaf in the image for subsequent CNN training; on the other hand, tropical disease detection studies in bananas using the random forest algorithm to accurately identify soil properties associated with disease symptoms reported accuracy values of around 85.4% [42–44]. Therefore, it is possible to effectively use the proposed architecture where the learning transferability of the model can be used to extend the scope in the long term.

However, crop disease detection has the following characteristics: the types of crops and diseases are complex and diverse [45], and the field images contain weeds, soil, and other factors, which lead to complex image backgrounds [46–48], the images are greatly affected by the shooting environment and conditions, the manifestations of different diseases are very different, and the various development processes of the same disease are very different.

In this paper, to solve the problems existing in crop disease detection, the specific contents are as follows: (1) propose an IASM mechanism to improve the accuracy and efficiency of the model, achieve model weight reduction through Ghostnet and WBF structure, and combine BiFPN and fast normalization fusion for weighted feature fusion to speed up the learning efficiency of each feature layer. The effectiveness of the optimization method is verified by ablation experiments. (2) Using data augmentation and other operations to create a peanut disease dataset, the transfer learning ability of the model is verified

by experiments. (3) The comparison test with other models shows that the optimized lightweight model can achieve fast and efficient detection and classification of diseases.

The main contents of the paper are as follows. Firstly, the relevant content and related work are introduced, and the research objectives are proposed. Then, the research object (dataset) and operation process, model principle and improvement, and model training and testing are introduced. The results are then analyzed to prove the feasibility and advancement of the model proposed in this study. Finally, the content of the full text is discussed and summarized, and the direction of future research is prospected.

## 3. Materials and Methods

### 3.1. Dataset Acquisition

In this paper on improving the accuracy and efficiency of the crop disease classification model, the application in the detection and classification of peanut diseases (brown spot and rust) was realized. The experiment used the PlantVillage dataset, which mainly contains images of 61 categories of 10 species. Since the dataset contained many cross-label data, the images containing the word "copy" in the file name were deleted to ensure that the dataset provided valid and usable information during the training process. The total number of training images and validation images were 31,718 and 4540, respectively, and the number of images in the test sets A and B were 4514 and 4513, respectively.

The method of the self-made dataset adopted in this study is as follows: first, the peanut brown spot and rust images are collected by a digital camera (Canon EOS 80D). Then, through the processing of image noise removal and the feature labeling operation of the dataset, a self-made peanut disease image dataset is produced. Finally, a total of 3265 peanut brown spot images and 1366 peanut rust images were obtained. The dataset is divided into training set, validation set, and test set according to the ratio of 8:1:1. The specific parameters of the final experimental dataset are shown in Table 2.

**Table 2.** The specific parameters of the dataset in this study (including the dataset of the model evaluation test and the transfer learning test).

| Datasets | Classes | Train Images | Validation Images | Test Images | Size (Pixels) |
|---|---|---|---|---|---|
| PlantVillage | 61 | 31,718 | 4540 | 4514/4513 | 256 × 256 |
| PlantDoc | 27 | 2098 | 221 | 279 | 640 × 640 |
| Peanut Brown Spot | - | 2608 | 326 | 331 | 256 × 256 |
| Peanut Rust | - | 1088 | 136 | 142 | 256 × 256 |

### 3.2. Image Preprocessing

To ensure that parameters such as image size meet the needs of model training and reduce noise during image acquisition, it is necessary to preprocess images and unify features, such as image size and color, which, in turn, can expand the number of datasets. The more commonly used image preprocessing methods include image vector normalization and data augmentation, such as resize, padding resize, and letterbox. The image transformation methods mainly include flip, blur operations, color enhancement, edge detection enhancement, logarithmic transformation, and image denoising. In this paper, the combination of the above methods was used to transform the collected images to increase the number of datasets, avoid the overfitting of the model, ensure the availability of the dataset, and improve the training speed of the model. An example of image preprocessing operations is shown in Figure 1.

The LabelImg tool, a graphic image annotation tool written in Python, was used for image calibration. Image annotation was performed in VOC format to obtain the xml and txt files of the training set and test set images, including image names, image size, target category name, target frame position, and other data information. The image data was converted into data information, and the data configuration file and model configuration

file were modified to facilitate the testing of the homemade peanut brown spot image dataset. The labeling process is shown in Figure 2a.

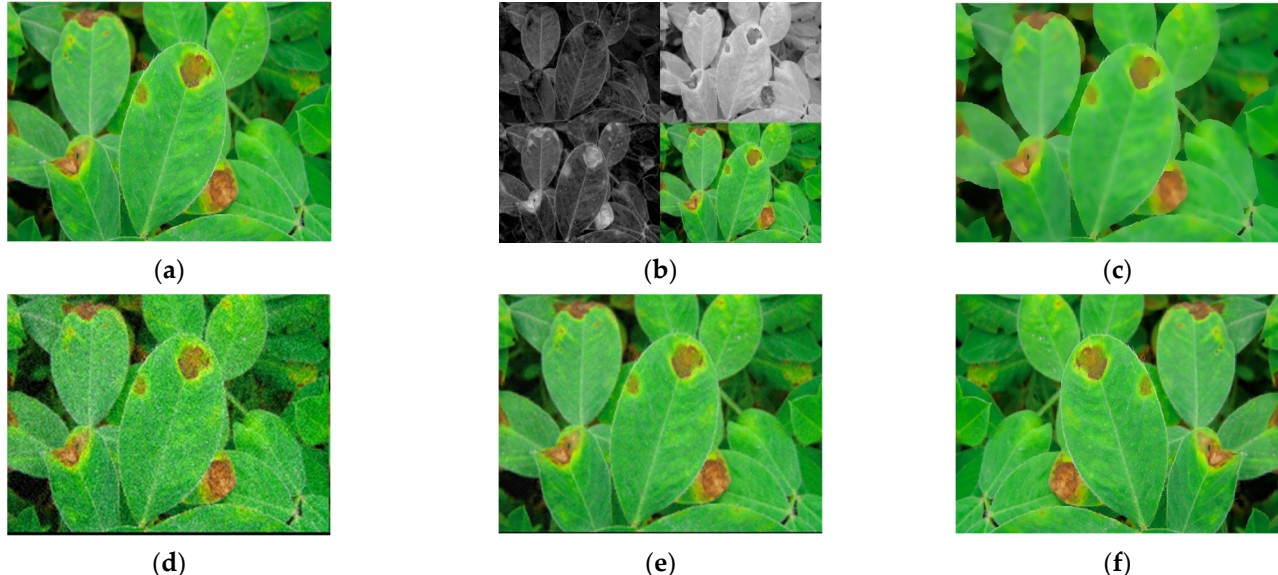

(**a**)      (**b**)      (**c**)

(**d**)      (**e**)      (**f**)

**Figure 1.** Example of image preprocessing operations (channel separation (**b**), mean shift filtering (**c**), noise detection (**d**), filter detection (**e**), and inversion (**f**) operations are performed on the original picture (**a**), respectively).

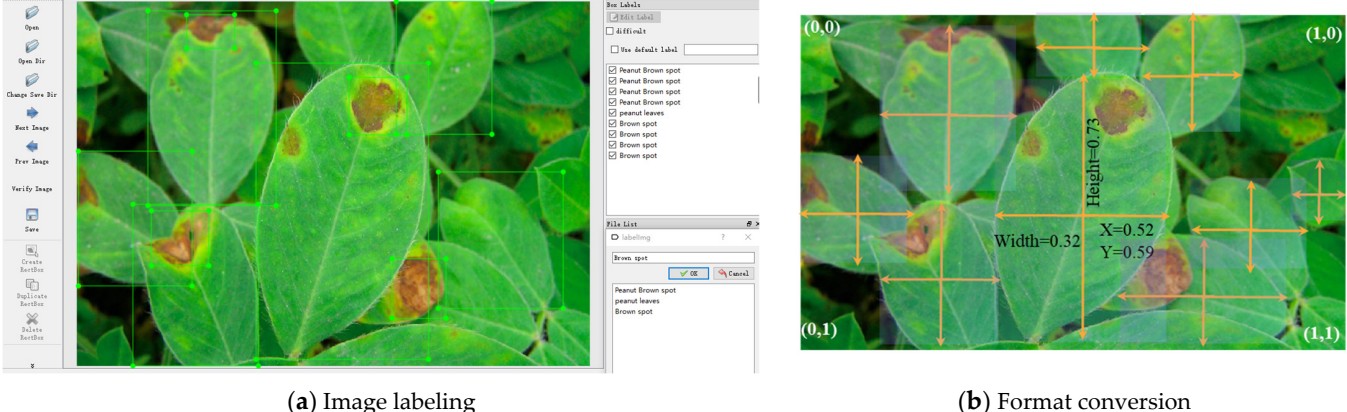

(**a**) Image labeling          (**b**) Format conversion

**Figure 2.** Example of image labeling (**a**) and format conversion (**b**).

The VOC format saved after labeling needs to be converted to YOLO format. The co-ordinate information saved in the VOC label data is the co-ordinate position of the target frame $[x_{min}, x_{max}, y_{min}, y_{max}]$, which represent the X and Y co-ordinate values of the upper left corner and lower right corner of the target frame, respectively. The YOLO label data contain: $[class, X\_center, Y\_center, width, height]$, which represent the category of the target box, the X and Y co-ordinates of the center point, and the width and height of the target frame, respectively. The conversion process equation between VOC format and YOLO format (take $x_{min}, x_{max}$ as an example) is as follows in Equation (1):

$$\begin{cases} x_{max} + x_{min} = 2x\_center \\ x_{max} - x_{min} = \omega * width \end{cases} \tag{1}$$

After finishing, Equation (2) can be obtained:

$$\begin{cases} x_{min} = x_{center} - \frac{1}{2} * \omega * width \\ x_{max} = x_{center} + \frac{1}{2} * \omega * width \end{cases} \tag{2}$$

where $x_{min}, x_{max}$ represent the $X$-co-ordinate values of the upper left corner and the lower right corner of the target frame, respectively.

### 3.3. Plant Disease Detection and Classification Model

#### 3.3.1. The Principle of the YOLOv5 Model

The process of the YOLOv5 network performing a recognition is equivalent to a regression calculation process: first, the input image is divided equally into $S * S$ grids; and then the image is sent into the network to predict whether there is a target, target category, and target bounding box in each grid; finally, the predicted bounding box is non-maximum suppression to be selected and output, and the output dimension is $S \times S \times (B \times 5 + C)$. The class-specific confidence score of the network represents the probability of the category appearing and the degree to which the predicted box matches the object, which can be obtained by Equation (3):

$$\begin{aligned} C_i^j &= P_r(Class_i|Object) * P_r(Object) * IOU_{pred}^{truth} \\ &= P_r(Class_i) * IOU_{pred}^{truth} \end{aligned} \tag{3}$$

where $C_i^j$ represents the confidence of the $j$-th bounding box of the $i$-th grid cell. $P_r(Class_i|Object)$ represents the probability that the target belongs to the $i$-th category; $P_r(Object)$ represents the probability of whether there is an object in the current box; $IOU_{pred}^{truth}$ represents the ratio of the intersection and union of the predicted box and the real box; and $P_r(Class_i)$ represents the probability that the $i$-th category appears.

The structure of the YOLOv5 network is mainly divided into four layers: the input layer, the backbone layer, the neck layer, and the prediction layer. The specific structure is shown in Figure 3.

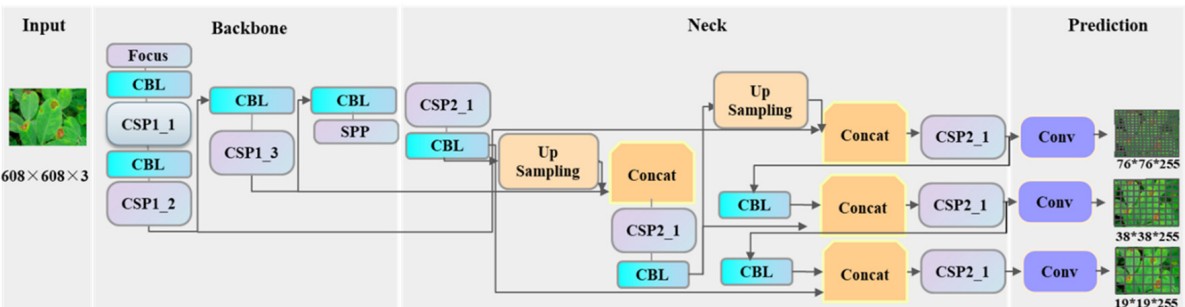

**Figure 3.** The structure of the YOLOv5 model (each layer and functional structure mainly include: input: mosaic data enhancement and adaptive anchor box calculation; backbone: focus structure and CSP structure; neck: FPN + PAN structure; prediction: GIOU_Loss).

(1) Input layer: based on the CutMix data enhancement method, the mosaic data enhancement method is proposed, which adaptively calculates the best anchor point frame according to the name of the dataset, and adaptively adds the least black border to the scaled image.

(2) Backbone layer: the structure of the Focus benchmark network, CSPDarknet53, and SPP is used. The CSPDarknet53 structure is used. The first layer (Focus layer): pixels are periodically extracted from high-resolution images and reconstructed into low-resolution images to reduce loss of original information and computational redundancy. The second layer (CSP structure): includes two types of CSP structure with Resunit ($2\times$CBL convolution + residual) and ordinary CBL CSP structure, which

is applied to the backbone layer and the neck layer, respectively. It is worth noting that this layer enhances the gradient value of backpropagation between layers and the generalization ability of the model by increasing the residual structure. The third layer (SPP structure): extract features of different scales through a pooling of different kernel sizes, and then perform feature fusion through stacking.

(3) Neck layer: the series FPN + PAN structure performs feature fusion and multi-scale prediction between different layers from bottom to top, strengthening the spread of semantic features and positioning information. The CBL module after the Concat operation is replaced by the CSP2_1 module, which helps to locate the pixels to form the mask.

(4) Prediction layer: the YOLOv5 model uses GIOU_Loss as the loss function of the bounding box, increases the intersection scale, and uses the weighted NMS method to screen the target box, which has a better classification effect for occluded overlapping targets.

3.3.2. Model Improvement

The YOLO algorithm has been researched in the recognition and classification of fruits [45,49] and marine organisms [50,51], but, due to the significant differences in the characteristics of different detection targets, the model needs to be improved and optimized to meet the detection and classification requirements. In view of the complex and diverse plant species and disease species in the disease detection task, the dataset background is complex and diverse, and the images are greatly affected by the shooting environment. However, the existing model has the limitation of too large a number of parameters and lack of attention to key areas. In order to improve the efficiency and accuracy of plant disease detection, identification, and classification, this study proposes an optimized plant disease detection and classification algorithm. The specific optimization work is as follows: (1) the improved attention submodule (IASM) is proposed to improve the accuracy and efficiency of the model; (2) the Ghostnet structure is introduced to reduce the amount of calculation, and the weighted frame is used to fuse WBF for postprocessing to realize the lightweight of the model; (3) the original FPN + PANet structure is replaced with the BiFPN structure, and Fast normalized fusion is used for weighted feature fusion to improve operation speed.

1. The improved attention submodule (IASM)

Just as when seeing an image for the first time, the brain pays general attention to the big picture of the image. The YOLOv5 model detects sub-regions of the entire image and lacks attention to key areas. If it is set in advance to find an object in the image (such as a dog), the brain will quickly scan the image, and, according to the previous impression of the dog, it will assign higher attention to similar parts of the image and focus on it. The process is the subconscious ability of the brain to find key areas. Researchers use this as the basis for model training and propose the Attention Mechanism. In order to improve the model's attention to the key areas of plant disease images, this paper proposes the Improved Attention Submodule (IASM), which acquires the target area that needs to be focused on by scanning the image globally and gives the area greater weight to obtain more details. At present, the most widely used attention mechanisms are mainly channeled attention, spatial attention, branch attention, and multi-class mixing mechanisms. Among them, channel attention uses a three-dimensional arrangement to save three-dimensional information and uses a multi-layer perceptron (MLP) to amplify the channel-space dependence, which can adaptively calibrate the channel weights. Spatial Attention first performs global average or maximum pooling on the channel, and then obtains attention from the spatial level. The structure is shown in Figures 4 and 5, respectively.

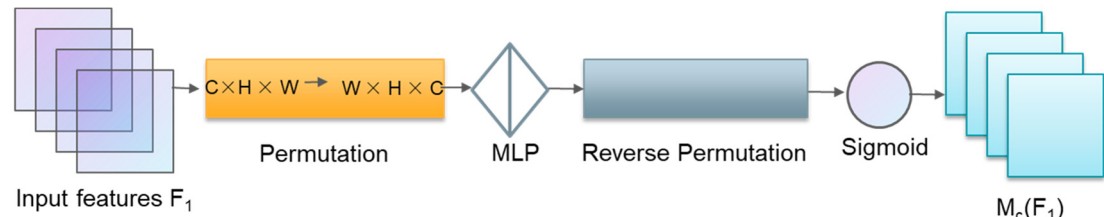

**Figure 4.** The structure of the channel attention submodule (contains MLP capable of amplifying channel-space dependencies).

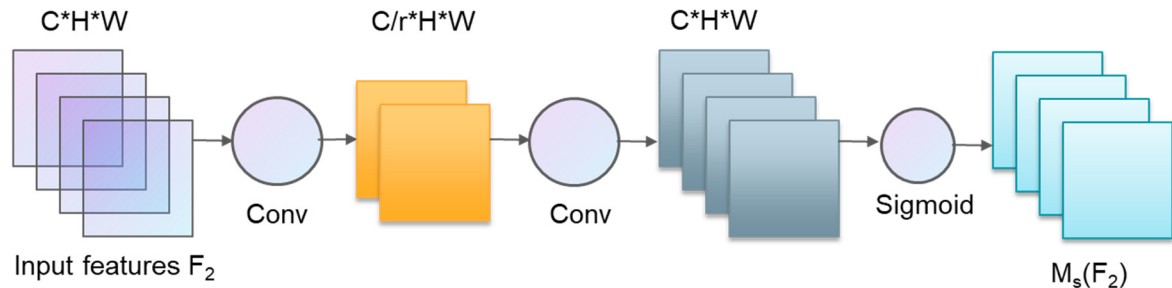

**Figure 5.** The structure of the spatial attention submodule (average pooling followed by max pooling).

The improved attention submodule preserves and fuses information by combining channel attention and spatial attention mechanisms, assuming the intermediate feature $F(H \times W \times C)$ as the output. The useful information (such as simple edges and shapes) is captured by each feature layer to obtain a more complex semantic representation of the input, expecting the network to pay more attention to the essential parts. Therefore, to enhance the discrimination of shallow semantic information, this study introduces the improved attention submodule for network optimization based on considering the 3D interaction, and adopts the structure of ResBlock + CBAM to optimize the model. Combined with the function of channel attention and spatial attention to pay attention to the key objects and key areas of the image, the operation speed of key target areas is enhanced, and the parallel and serial performance comparisons of channel attention and spatial attention are carried out, respectively. The results show that the optimization effect of the model is the best by adopting parallel and spatial attention followed by channel attention. The structure optimized in this paper is shown in Figure 6.

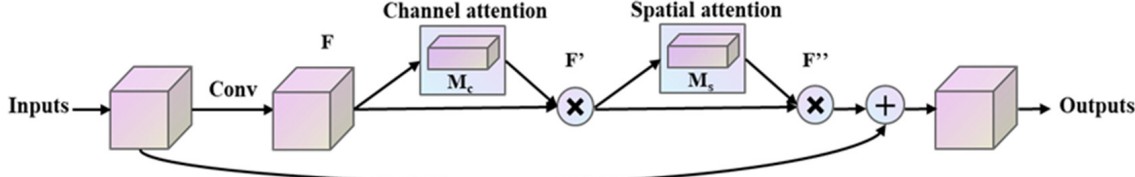

**Figure 6.** The improved attention submodule (IASM) (structure of serial channel attention and spatial attention).

The main process is to obtain the intermediate feature $F(H \times W \times C)$ by performing the convolution operation and the average and maximum pooling operations on the inputs, and then use the activation function to calculate the output channel attention *Mc*, which is multiplied by the MLP to obtain the channel attention. Force distribution map $F'$, illustrates the importance of channels in the feature map. The channel attention map $F'$ is averaged, max pooled, and convolutional to obtain the spatial attention *Ms*, *Ms*, and $F'$, which are multiplied to obtain the spatial attention map $F''$, and the channel attention map is related to the spatial attention map. This is multiplied and added to the original input to obtain the final output. The calculation process of channel attention mainly includes attention distribution and weighted average. The attention distribution is to calculate the

attention distribution of all input information and input the task-related information into the network to save computing resources, introduce the query vector q, and calculate the attention distribution $\alpha_i$ through the scoring function $S(x_i, q)$, namely:

$$\alpha_i = p(z = i|X, q) = softmax(S(x_i, q)) \tag{4}$$

where the attention distribution $\alpha_i$ represents the degree of attention to the *i*-th input vector when performing the detection task.

Weighted distribution $att(X, q)$ is a process of useful weighting information and the weighted average of all input attention. The calculation process is as follows:

$$att(X, q) = \sum_{i=1}^{N} \alpha_i X_i \tag{5}$$

where the attention distribution $\alpha_i$ represents the degree of attention to the *i*-th input vector when performing the classification task; and $X_i$ represents the weight assigned to the *i*-th input vector.

To accurately locate the position of disease and improve the characterization ability of the detection and classification model, an improved process of weight fusion applicable to plant disease classification is proposed based on the introduction of the improved attention submodule and the idea of feature rescaling and fusion in the SENet module. The module structure is shown in Figure 7. The input vectors are subjected to maximum global pooling and average pooling operations, and then added after the operations of fully connected layer 1 (FC_1), modified linear unit, and fully connected layer 2 (FC_2). After the result is cross-multiplied with the original input, the result is obtained and output to the next layer. In the process of maximum pooling, first, the maximum possible value of each candidate box is calculated and used as the main influencing factor of the result. Then, the IOU of the candidate frame corresponding to the maximum possible value is calculated, which is the basis for measuring the accuracy of the prediction model.

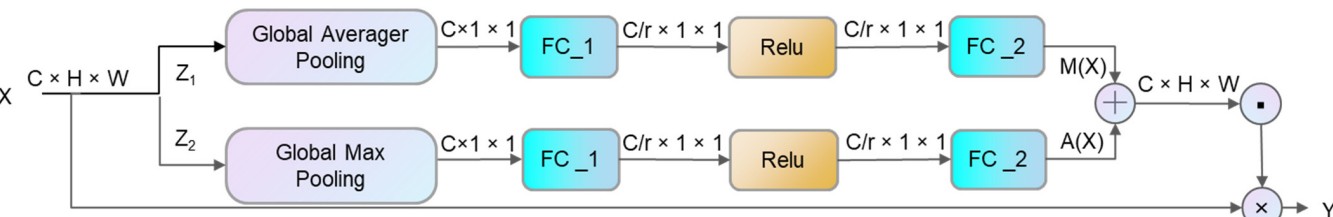

**Figure 7.** The process of weight fusion (the input vector X is subjected to global max pooling and global average pooling, respectively, and then added through the FC_1, Relu, and FC_2 layers, respectively. The result is cross-multiplied with the original input X to obtain the final output vector Y).

The attention mechanism is used to generate the weights of different connections dynamically. Assuming that the input sequence is $X = \{x_1, x_2 \ldots \ldots, x_N\} \in R^{d1 \times N}$, and the output sequence is $H = \{h_1, h_2 \ldots \ldots, h_N\} \in R^{d2 \times N}$, three sets of vector sequences can be obtained by linear transformation:

$$Q = W_Q X \in R^{d1*N} \tag{6}$$

$$K = W_K X \in R^{d2*N} \tag{7}$$

$$V = W_V X \in R^{d3*N} \tag{8}$$

where $Q$, $K$, and $V$ are the query vector sequence, the key vector sequence, and the value vector sequence, respectively. The above formulas represent the learnable parameter matrix, respectively, and the output vector $h_i$ can be obtained by Equation (9):

$$h_i = att((K, V), q) = \sum_{i=1}^{N} \alpha_{ij} V_j \tag{9}$$

where $i, j \in [1, N]$, represent the positions of the output and input vector sequences, respectively; and $\alpha_{ij}$ represents the connection weight, which is dynamically generated by an optimized global attention mechanism.

2.　　Ghostnet

Since the feature map layer in the plant disease image dataset selected in this study contains more redundant information, the size of the model is also increased when the validity of the input data is increased. To reduce the calculation amount of the model and further improve the operation speed of the model, this study proposes to combine Ghostnet to reduce the weight of the network, and use a low-cost calculation method to obtain redundant feature map layer information. The convolution operation is divided into two steps: small-scale convolution and channel-by-channel convolution. Small-scale convolution realizes the degree of parameter optimization by customizing the number of convolution kernels. Channel-by-channel convolution combines original convolution with simple linear transformation to get more features. Ghostnet adopts identity and linear transformation at the same time, and realizes the lightweight of the model based on maintaining the original features. The model structure is shown in Figure 8.

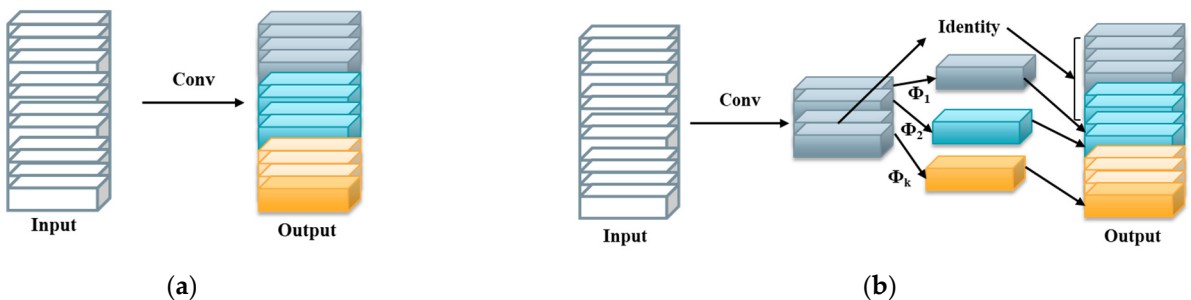

(**a**)　　　　　　　　　　　　　　　　　　　　　　　　　(**b**)

**Figure 8.** Structure of conventional convolution model (contains only conventional convolution operations) (**a**) and Ghostnet model (convolution operations include a small amount of convolution and channel-by-channel convolution) (**b**).

The calculation process of conventional convolution is as in Equation (10); the input data $X \in R^{c \times h \times w}$ pass through the convolution kernel $f \in R^{c \times k \times k \times n}$, and, finally, obtains the output $Y \in R^{h' \times w' \times n}$. The calculation amount reaches $n \cdot h' \cdot w' \cdot c \cdot k \cdot k$, and the number is huge.

$$Y = X * f + b \tag{10}$$

The Ghostnet model obtains a part of the output $Y' \in R^{h' \times w' \times m}$ by customizing the number of convolution kernels $m, (m \ll n)$, while the rest of the s-dimensional feature output $y_{ij}$ is generated by a linear calculation, which greatly simplifies the calculation.

$$Y' = X * f' \tag{11}$$

$$y_{ij} = \phi_{i,j}(y_i'), \forall i = 1, \ldots, m, j = 1, \ldots, s \tag{12}$$

where $\phi_{i,j}$ is the linear transformation function for generating the $j$-th feature map of $y_i'$.

Some plant disease images contain multiple target types, that is, an image often contains multiple target boxes, which are likely to overlap and misjudgment occurs. Therefore, it is necessary to optimize the output process of the bounding box of the model. The non-

maximum suppression method NMS (non-maximum) of the original model simply divides the pros and cons of the prediction box by deleting operations. In this study, by adding the bounding box fusion algorithm weighted boxes fusion (WBF) to the original model, the boundaries are sorted according to the confidence score and Equation (13) is used to calculate the co-ordinates and confidence score of the target box, which are determined by the new confidence score. Weights are used for co-ordinate weighting. This is equivalent to giving high-confidence target boxes a higher priority for the output, and so on. Assuming that the original model predicts two close target frames: A: $[Ax_1, Ay_1, Ax_2, Ay_1, As]$, B: $[Bx_1, By_1, Bx_2, By_1, Bs]$, where $(x_1, y_1), (x_2, y_2)$ represent the co-ordinates of the upper left corner and the lower right corner of the target frame, respectively, and $s$ is the confidence score of the target frame, then the target frame C $[Cx_1, Cy_1, Cx_2, Cy_1, Cs]$ is finally obtained through the fusion of A and B. The calculation process is as follows:

$$
\begin{aligned}
C_{x1} &= \frac{Ax_1 \times As + Bx_1 \times Bs}{As + Bs} \\
C_{x2} &= \frac{Ax_2 \times As + Bx_2 \times Bs}{As + Bs} \\
C_{y1} &= \frac{Ay_1 \times As + By_1 \times Bs}{As + Bs} \\
C_{y2} &= \frac{Ay_2 \times As + By_2 \times Bs}{As + Bs} \\
Cs &= \frac{As + Bs}{2}
\end{aligned}
\tag{13}
$$

Through this improved method, the confidence score of the target frame is used as a weight to determine the contribution of the original frame to the fusion frame, so that the position and size of the fusion frame are closer to the frame with high confidence, ensuring the accuracy of positioning and classification.

3.　　The Bidirectional Feature Pyramid Network

The neck layer of the YOLOv5 network adopts the FPN + PANet structure for feature aggregation. Due to the one-way structure of FPN, PANet adds a bottom-up inter-layer transmission path, which improves the connection between high-level semantic information and bottom-level location information. It is beneficial to the classification and localization of the target, but it consumes extra time due to the establishment of one more path. The repeated weighted bidirectional feature pyramid network (BiFPN) ignores nodes with only one input edge, that is, the network believes that if a node has only one input edge, the node's contribution to the feature network is small.

However, since different input features have different resolutions, different scale feature layers have different contributions. Therefore, this paper uses the BiFPN structure to replace the original FPN + PANet structure, and the improved network structure is shown in Figure 9. In addition, by setting the learning parameters to ensure the consistency of the input layer size, according to the importance of different feature layers, additional weights are added to each feature layer, and fast normalized fusion (Fast normalized fusion) is used to carry out feature layer weighted features. The fusion process is as follows:

$$
O = \sum_i \frac{w_i * I_i}{\epsilon + \sum_j w_j}
\tag{14}
$$

$$
P_6^{td} = Conv\left(\frac{w_1 * P_6^{in} + w_2 * Resize(P_7^{in})}{\epsilon + w_1 + w_2}\right)
\tag{15}
$$

$$
P_6^{out} = Conv\left(\frac{w_1' * P_6^{in} + w_2' * P_6^{td} + w_3' * Resize(P_5^{out})}{\epsilon + w_1' + w_2' + w_3'}\right)
\tag{16}
$$

where *Resize* is the up-sampling or down-sampling operation. $\omega$ denotes the weight parameter that determines the significance of additional features.

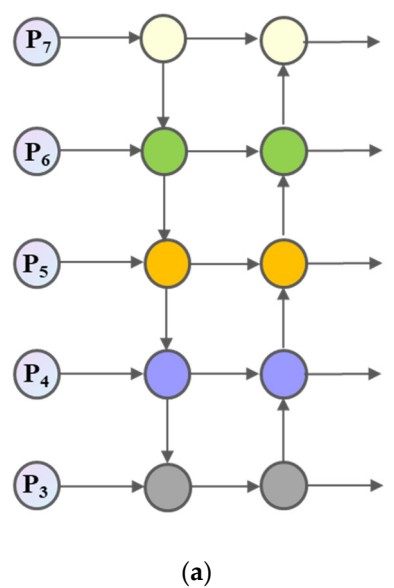
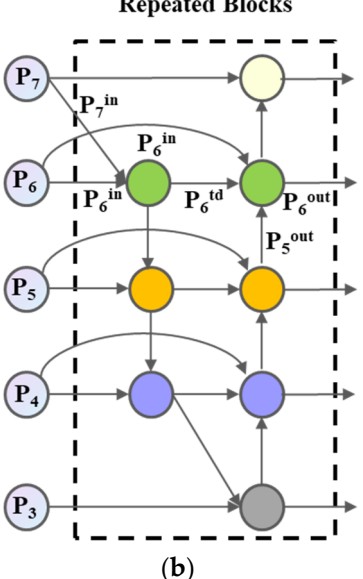

**Figure 9.** Structure of PANet (bottom-up secondary fusion) (**a**) and BiFPN (edges with contextual information added to the original FPN module) (**b**).

After the above improvements, the improved optimized lightweight YOLOv5 network model of this study is finally proposed. The specific structure is shown in Figure 10.

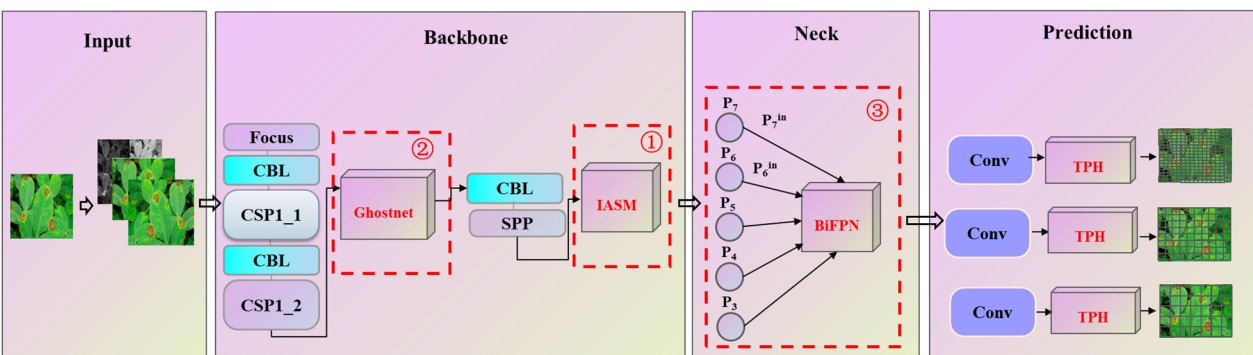

**Figure 10.** The structure of the optimized lightweight YOLOv5 model proposed in this study (including the three proposed improvements). (Added IASM (①), Ghostnet (②) and BiFPN (③) to the original model).

### 3.3.3. Performance Indicators

The speed and accuracy of the object detection algorithm are the main indicators to measure the algorithm's performance. Therefore, this paper evaluated the performance of the improved algorithm by comparing the performance indicators, such as accuracy rate, recall rate, operation time, and loss function.

1. Accuracy and Precision

The classification results are given by the algorithm usually include four cases: *T* and *F*, respectively, indicating that the model prediction is true or false. *P* and *N* indicate that the instance is predicted to be a positive or negative class. The combination of the prediction categories constitutes all the prediction results, and the performance of the model can be evaluated according to the ratio between the different prediction results.

$$Accuracy = \frac{TP + TN}{TP + TN + FP + FN} \times 100\% \tag{17}$$

$$Precision = \frac{TP}{TP + FP} \times 100\% \tag{18}$$

$$Recall = \frac{TP}{TP + FN} \times 100\% \tag{19}$$

$$F1 = \frac{2(Precision \times Specificity)}{Precision + Specificity} \times 100\% \tag{20}$$

where Accuracy (*A*) refers to the ratio of correct targets to the total number of targets. Precision (*P*) refers to the ratio of the number of correct objects detected to the number of correct objects in the sample. Recall (*R*) refers to the ratio of the number of correct objects detected to the number of objects in the sample. Therefore, in optimizing the algorithm, it is necessary to balance the relationship between *A*, *P*, and *R*. To comprehensively evaluate the algorithm's performance, the *F1* value is introduced. The *F1* value is the weighted harmonic average of *P* and *R*. The test method is more effective when the *F1* value is high.

2. Loss function

The YOLOv5 model uses the mean square error between the output image vector and the real image vector as a loss function, including confidence loss, classification loss, and localization box. The weighted addition of the three parts obtains the overall loss, and the attention to the relevant loss can be adjusted by assigning weights.

The confidence loss ($l_{obj}$) includes the confidence prediction with the object and the confident prediction of the box without an object in the box, which is defined as follows:

$$l_{obj} = \lambda_{obj} \sum_{i=0}^{S^2} \sum_{j=0}^{B} \int_{ij}^{obj} (C_i - \hat{C}_i)^2 + \lambda_{noobj} \sum_{i=0}^{S^2} \sum_{j=0}^{B} \int_{ij}^{noobj} (C_i - \hat{C}_i)^2 \tag{21}$$

The classification loss ($l_{cls}$) determines whether there is an object center in the *i*-th grid, which is defined as follows:

$$l_{cls} = \lambda_{class} \sum_{i=0}^{S^2} \sum_{j=0}^{B} \int_{i,j}^{obj} \sum_{c \in classes} (p_i(c) \log(\hat{p}_i(c))) \tag{22}$$

The localization loss ($l_{box}$) includes the center point co-ordinate ($x$, $y$) error and the width and height co-ordinate ($w$, $h$) error of the target frame and the prediction frame, which are defined as follows:

$$l_{box} = \lambda_{coord} \sum_{i=0}^{S^2} \sum_{j=0}^{B} \frac{\int_{ij}^{obj} (2 - w_i \times h_i)}{\left[ (x_i - \hat{x}_i)^2 + (y_i - \hat{y}_i)^2 + (\omega_i - \hat{\omega}_i)^2 + \left( h_i - \hat{h}_i \right)^2 \right]} \tag{23}$$

In summary, the loss function is defined as follows:

$$Loss = l_{obj} + l_{cls} + l_{box} \tag{24}$$

$$Loss = \lambda_{coord} \sum_{i=0}^{S^2} \sum_{j=0}^{B} \int_{i,j}^{obj} (x_i - \hat{x}_i)^2 + (y_i - \hat{y}_i)^2 + \lambda_{coord} \sum_{i=0}^{S^2} \sum_{j=0}^{B} \int_{i,j}^{obj} (\sqrt{\omega_i} - \sqrt{\hat{\omega}_i})^2 + \left( \sqrt{h_i} - \sqrt{\hat{h}_i} \right)^2$$
$$+ \sum_{i=0}^{S^2} \sum_{j=0}^{B} \int_{i,j}^{obj} (C_i - C_{ii})^2 + \lambda_{noobj} \sum_{i=0}^{S^2} \sum_{j=0}^{B} \int_{i,j}^{noobj} (C_i - \hat{C}_i)^2 + \sum_{i=0}^{S^2} \int_{i,j}^{obj} \sum_{c \in classes} (p_i(c) - \widehat{p_i(c)})^2 \tag{25}$$

where $\int_{i,j}^{obj}$ represents judging whether the j-th box in the i-th grid is responsible for this object. $S^2$ is the grid number. *B* is the number of prediction boxes in the grid and *C* is the prediction confidence. $C_i$ is the true value of confidence. $C_{ii}$ is the diagonal distance

between the predicted box and the ground-truth box. $p_i(c)$ is the probability of being predicted to be the true category $c$.

### 3.4. Model Training and Testing

The dataset in this study is divided into training set, validation set, and test set in a ratio of 8:1:1. The different network models: Faster R-CNN, VGG16, SSD, EfficientDet, YOLOv4, YOLOv5, and the improved YOLOv5 model were selected for training and testing. Adam was used as the optimizer with a learning rate of 0.0001. The specific operating environment parameters are shown in Table 3.

**Table 3.** Operating environment parameters.

| Index | Parameter |
|---|---|
| RAM | 8 G |
| CPU | Intel®Core™i5-7200 CPU@2.50 GHz |
| GPU | RTX 2070 (8 GB) |
| CUDA | 4.10.1 |
| Development Environment | Pytorch1.4 |
| Programing Language | Python1.8 |
| Model | YOLOv5 |

The YOLOv5 model includes four different structures [18]: YOLOv5s (the smallest), YOLOv5m, YOLOv5l, and YOLOv5x (the largest). The comparison of parameters is shown in Table 4. The F1 of YOLOv5s was relatively lower (F1 was 0.873, which was 1.8% lower than YOLOv5x). However, due to the small number of network layers and parameters and small memory footprint, YOLOv5s improved the running speed (the operation time was 2.2 ms, which was 63.3% faster than YOLOv5x). Therefore, YOLOv5s has the advantages of fast speed, small size, and high accuracy, which meets the requirements of this study for operation speed and accuracy. Therefore, this study optimized and improved the structure of the YOLOv5s model and proposed a detection and classification method suitable for plant diseases.

**Table 4.** Comparison of different versions of the model parameters of YOLO v5 (general evaluation indicators, such as speed and FLOPs of different versions of the model).

| Model | Layers | mAP | F1 | Speed (ms) | FPS | Params (M) | FLOPs (B) |
|---|---|---|---|---|---|---|---|
| YOLO v5s | 191 | 14.08 | 0.873 | 2.2 | 455 | 7.3 | 17.0 |
| YOLO v5m | 263 | 41.33 | 0.883 | 2.9 | 345 | 21.4 | 51.3 |
| YOLO v5l | 335 | 90.85 | 0.887 | 3.8 | 264 | 47.0 | 115.4 |
| YOLO v5x | 407 | 169.29 | 0.889 | 6.0 | 167 | 87.7 | 218.8 |

In this study, the YOLOv5s model is selected as the original model. The test process is as follows. First, a new folder is created in the model folder for storing the images marked with LabelImg and the contents of the marked images. Then, the dataset is divided by the Split function, and the ratio of training set, validation set, and test set is adjusted. The ratio of the dataset set in this study is 8:1:1. In addition, format conversion is performed according to the method described in Section 3.2, obtaining the YOLO format file of the dataset, loading the training configuration file, importing the model configuration file, and modifying the model parameters according to the relevant configuration to improve the model. Then, according to the target category name and number of the dataset, the configuration file is modified. In terms of generating a priori frame, the method of automatically obtaining anchors is adopted, and the Best Possible Recall (BPR) is detected and calculated through the clustering principle to determine whether to recalculate the anchors (when BPR < 0.98, clustering through k-means (Euclidean distance) is used to obtain new anchors; and when BPR > 0.98, no need to regenerate anchors).

In terms of model improvement, functional programs, such as the attention mechanism (IASM) proposed in this study, are added to the functional file of the model. In this study, the attention mechanism is added to the last layer of the BackBone of the model, and the features extracted by channel attention are added. As the input of spatial attention, and operations such as max pooling, average pooling, and dot product are performed, the final output vector is obtained. Then the numbers and coefficients of the subsequent layers of the model are modified accordingly to ensure the stability of the model function. The original convolution is divided into two steps. A small amount of convolution in the first step and a linear change in the second step are used to generate the remaining feature maps, and the output channels of the two steps are combined to obtain the final output.

After the configuration file of the model is modified, the training and parameter optimization of the model are started. First, the pretrained model and model weights file are downloaded to set the initial parameters of the model. The Adam optimization function is taken to optimize the model. The model is trained through the warm-up mechanism, so that the model is slowly stabilized within the initial epochs or steps and the learning rate is small, and then the preset learning rate of 0.0001 is selected for training, the model parameters updated, and the convergence of the model speed sped up, maintaining model stability. The optimization process is as follows:

$$W\_data = W\_data + W\_grad * lr \tag{26}$$

where $W\_data$ is the model parameter, $W\_grad$ is the parameter gradient, and $lr$ is the learning rate.

The convolution (Conv) process calls the autopod function to calculate the amount of padding required by the same padding. The default activation function SiLU was used, and the function form was as follows:

$$f(x) = x \cdot \sigma(x) \tag{27}$$

where $x$ is the input value, and $\sigma(x)$ is the SiLU function.

The model parameters are subjected to L2 regularization weight attenuation processing to reduce the output weight and prevent the model from overfitting.

## 4. Results and Analysis

In this paper, network models with different structures were selected and tested: Faster R-CNN, VGG16, SSD, EfficientDet, YOLOv4, YOLOv5, optimized lightweight YOLOv5, and other network models.

### 4.1. Model Parameter Optimization

The postprocessing process of YOLOv5 included two mappings, and the operation process was as follows:

$$\begin{aligned} b_x &= 2 * \sigma(t_x) - 0.5 + c_x \\ b_y &= 2 * \sigma(t_y) - 0.5 + c_y \\ b_w &= p_w(2 * (t_w))^2 \\ b_h &= p_h(2 * (t_h))^2 \end{aligned} \tag{28}$$

where $b$ is the size of the prediction box, including $[b_x, b_y, b_h, b_w]$, which represents the x-co-ordinate, y-co-ordinate, height, and width of the center point, respectively. $c_x$ and $c_y$ are the side lengths of the cell. $p_w$ and $p_h$ are the width and height of the prior box. $\sigma$ is the sig activation function, as shown in Figure 11.

In this study, Wandb was configured to visualize the training process and dynamically monitor the training status and operation of the model to observe the effect of training times on model performance and equipment conditions. The results are shown in Figure 12. In the process of model iteration 0–300 times, the model's parameters fluctuated greatly. In the process of model iteration 300–600 times, the model performance was continuously

optimized as the number of iterations increased. In the process of 900–1000 iterations of the model, the index gradually became stable, and the precision (*P*) reached about 93% and gradually stabilized. Therefore, considering the influence of the number of iterations on the stability and efficiency of the model, the optimal number of iterations in this study was 1000 (the number of iterations in subsequent experiments was 1000). After the model was trained 1000 times, the learning rate dropped by 0.1.

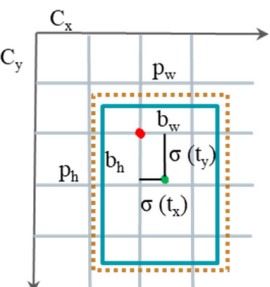

**Figure 11.** Schematic diagram of the calculation of the size of the prediction box.

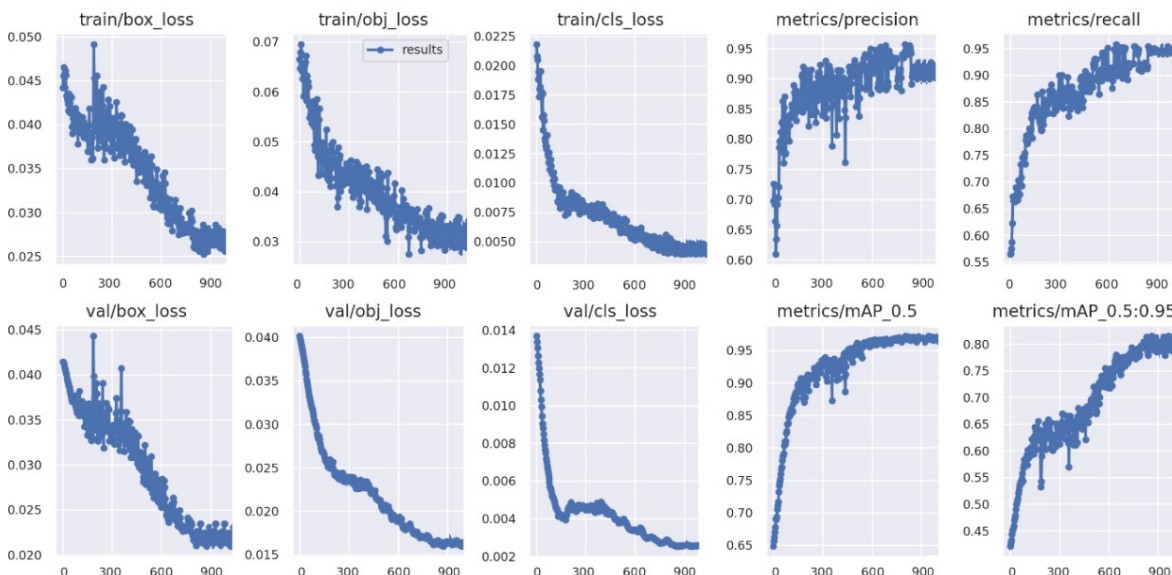

**Figure 12.** Visual analysis of model evaluation indicators during training (model iterations 0–1000 times).

### 4.2. Analysis of Test Results

The test results are shown in Figure 13. The performance of the model for crop disease classification was evaluated by comparing the operation time, loss function (*Loss*), precision (*P*), recall (*R*), F1 score (*F1*), and other indicators. The results are shown in Table 5. By analyzing the results, we can see that: (1) the size of the optimized lightweight model was 17.8 MB; compared with the Faster R-CNN, VGG, SSD, EfficientDet, YOLOv4 models, and YOLOv5, it saved 151.2 MB, 104.2 MB, 76.7 MB, 38.2 MB, 19.9 MB, and 0.8 MB, respectively, indicating that the optimized model greatly saved memory usage. (2) The average operation time of a single image was 15 ms, which was 93.6%, 90.5%, 81.0%, 53.1%, 37.5%, and 11.8% higher than the Faster R-CNN, VGG16, SSD, EfficientDet, YOLOv4, and YOLOv5 models, respectively, indicating that the optimized model improved the operation speed. (3) The precision, recall, and *F1* score of the optimized YOLO v5 model for disease classification were 93.73%, 92.94%, and 92.97%, respectively, which were 3.98%, 2.69%, and 2.29% higher than the original YOLOv5 model, indicating that the optimized model had better classification ability. Based on the above analysis, the method proposed in this study

had obvious advantages in model size, speed, and precision, which can meet the needs of plant disease detection and classification.

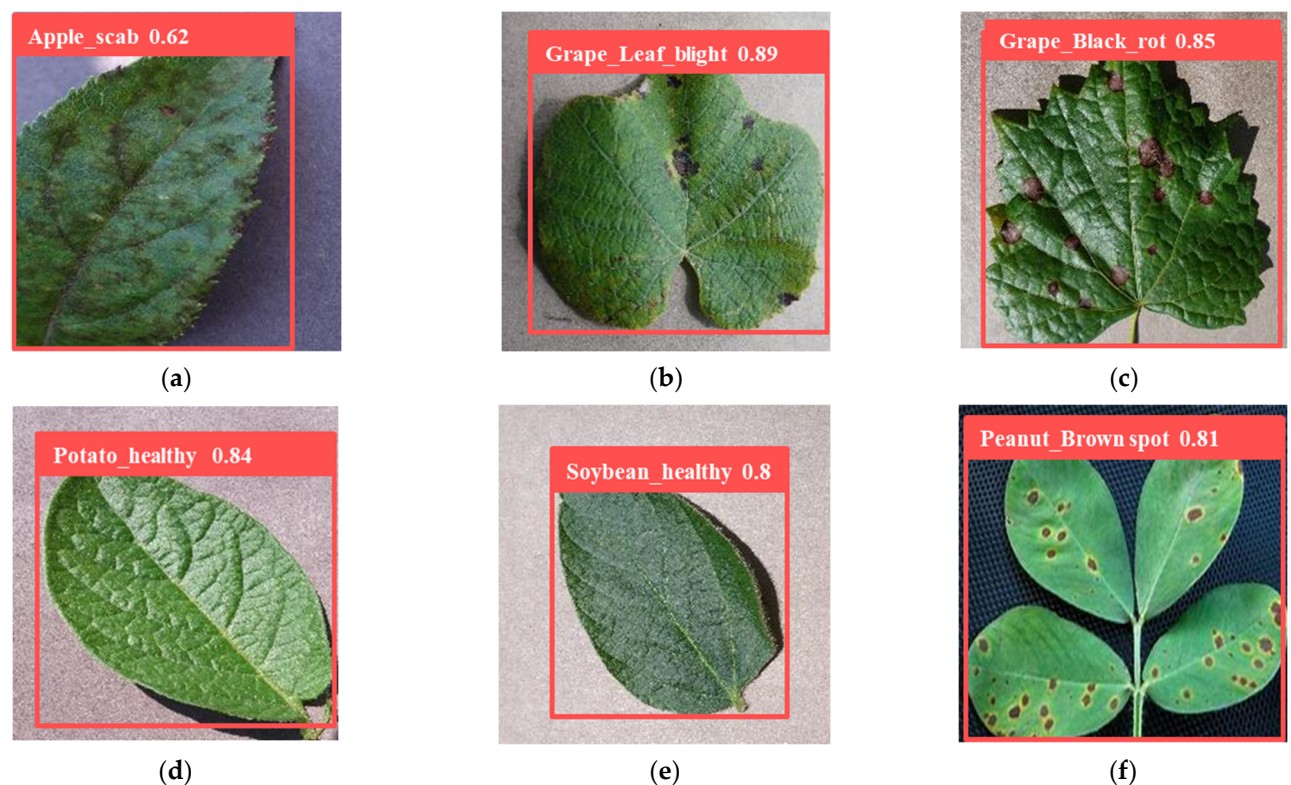

**Figure 13.** Partial example images of the recognition results of the improved YOLOv5 model (category name and confidence for classification). Representing the disease type (confidence) are Apple_scab (0.62) (**a**); Grape_Leaf blight (0.89) (**b**); Grape_Black rot (0.85) (**c**); Potato_healthy (0.84) (**d**); Soybean_healthy (0.8) (**e**); Peanut_Brown spot (0.81) (**f**), respectively.

**Table 5.** Comparison of recognition performance of different network models (currently mainstream single-stage and two-stage target detection models and the model proposed in this paper).

| Model | Model Size/MB | Iteration Number | Operation Time/ms | Loss | P/% | R/% | F1/% |
|---|---|---|---|---|---|---|---|
| Faster R-CNN | 165.0 | 1000 | 2.35 | 1.28 | 84.23 | 85.94 | 85.92 |
| VGG16 | 118.0 | 1000 | 1.58 | 1.56 | 86.66 | 85.85 | 84.18 |
| SSD | 90.5 | 1000 | 0.79 | 1.28 | 85.37 | 85.29 | 86.32 |
| EfficientDet | 52.0 | 1000 | 0.32 | 1.13 | 90.39 | 90.65 | 90.89 |
| YOLOv4 | 33.7 | 1000 | 0.24 | 1.04 | 86.59 | 87.20 | 87.74 |
| YOLOv5 | 14.6 | 1000 | 0.17 | 0.89 | 89.75 | 90.25 | 90.68 |
| Optimized YOLOv5 | 13.8 | 1000 | 0.15 | 0.56 | 93.73 | 92.94 | 92.97 |

To compare and analyze the process of crop disease identification by the optimized YOLOv5 model proposed in this study, the visual map of the features extracted by the convolutional layer of the model was output. In the experiment process, the self-attention model was used alternately with the convolutional layer and the recurrent layer, and the position encoding information was added to correct the connection weight to ensure the accuracy of the input information location. The basic features of the SAR image were extracted in the bottom-up forward path, and the outputs of the conv4_3, conv5_3, and conv7 convolutional layers of the network were extracted for fusion. The spatial size was selected 8, 16, and 31 times of the input image down-sampling, respectively. The three-layer feature maps were sent to the anchor point refinement module to eliminate the negative sample anchor point frame and correct the position of the positive sample anchor point

frame. The Feat_3 feature map was constructed using a multi-layer convolutional transfer layer, and the size is defined as 10 × 10 pixels.

From Figure 14, we can see that the features extracted by different feature maps had different emphases: the shallow feature maps retained all the input image features except for color. With the increase in the number of convolutional layers, the feature information of the deep feature image was gradually blurred, the texture features of the disease were steadily enhanced, and the extracted features were smaller. When the convolutional layer reached the maximum, the feature map contained only disease features for classification. By comparing the original network map (Figure 14a) and the output feature map of the optimized model (Figure 14b), we can see that when the color of the background is similar to the color of the disease, the original model extracted the background as the disease area, and the ability to locate and extract features was poor. The optimized model could distinguish disease features and background features through convolution kernel and attention mechanism and realize the extraction and location of disease features. Therefore, the enhanced model proposed in this study had a better ability to distinguish the background and disease of the image.

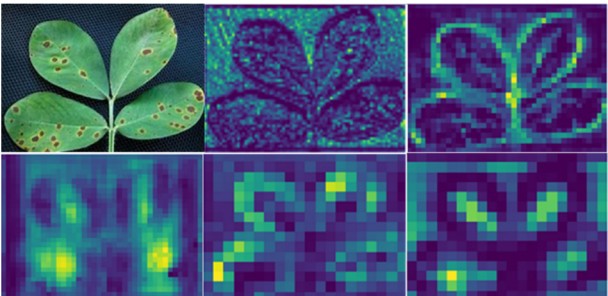
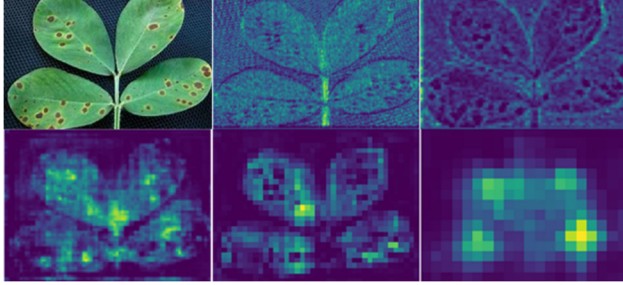

(**a**) Extracted feature map of original model      (**b**) Extracted feature map of optimized model

**Figure 14.** Extracted feature map of original model (**a**) and the optimized model (**b**).

### 4.3. Ablation Experiment

The experimental results of Section 4.2 showed that the improvement and optimization of the network using the fusion improved attention submodule (IASM), the BiFPN structure, and the Ghostnet structure had achieved relatively good experimental results. To verify the influence of the improvement methods on the results, the original YOLOv5 network was improved only through the improved attention submodule (IASM + YOLOv5), and only through the BiFPN structure (BiFPN + YOLOv5), and by combining the IASM + BiFPN structure (optimized YOLOv5). In addition, to verify the effectiveness of the IASM mechanism introduced in this paper, the channel attention mechanism (CAM + YOLOv5) and the spatial attention mechanism (SAM + YOLOv5) were introduced in the ablation experiment to improve the model structure. To reduce the impact of the dataset on the model performance, the same set is input to the improved network, and the results are shown in Table 6. By analyzing the results, we can see that: (1) the IASM +YOLOv5 model and the BiFPN + YOLOv5 model had a precision of 92.56% and 91.8%, which were 2.81% and 2.05% higher than the original model, respectively. The recall of the IASM + YOLOv5 model was 92.68%, which was 2.4% higher than the original model, indicating that the operation speed of the model was improved. (2) The CAM + YOLOv5 model improved the ability to transmit feature information, but the operation speed was significantly reduced (operation time was 19 ms, lower than 17 ms for the SAM + YOLOv5 model and 16 ms for the IASM + YOLOv5 model). The SAM + YOLOv5 model enhanced the utilization of spatial location information, but the loss function was improved (loss was 0.76, higher than 0.69 for the CAM + YOLOv5 model and 0.65 for the IASM + YOLOv5 model). The IASM + YOLOv5 model improves the speed and precision of the model at the same time (operation time was 0.16 s and precision was 91.80%). (3) The optimized YOLOv5 disease detection and classification model proposed in this paper had the best operation perfor-

mance, the F1 score reached 92.97%, and the comprehensive performance of the model was the best.

**Table 6.** The results of the ablation experiment (the improved methods proposed in this study and other similar methods are added to the model, respectively).

| Model | Iteration Number | Learning Rate | Operation Time/s | Loss | P/% | R/% | F1/% |
|---|---|---|---|---|---|---|---|
| YOLOv5 | 1000 | 0.0001 | 0.18 | 0.89 | 89.75 | 90.25 | 91.68 |
| IASM + YOLOv5 | 1000 | 0.0001 | 0.16 | 0.65 | 91.80 | 92.68 | 92.38 |
| BiFPN + YOLOv5 | 1000 | 0.0001 | 0.17 | 0.62 | 92.56 | 90.98 | 92.07 |
| WBF + YOLOv5 | 1000 | 0.0001 | 0.15 | 0.79 | 91.31 | 91.78 | 91.50 |
| CAM + YOLOv5 | 1000 | 0.0001 | 0.19 | 0.69 | 89.98 | 88.91 | 91.87 |
| SAM + YOLOv5 | 1000 | 0.0001 | 0.17 | 0.76 | 90.59 | 90.63 | 92.01 |
| NMS + YOLOv5 | 1000 | 0.0001 | 0.17 | 0.82 | 90.57 | 90.85 | 90.97 |
| Optimized YOLOv5 | 1000 | 0.0001 | 0.15 | 0.56 | 93.73 | 92.94 | 92.97 |

*4.4. Transfer Learning Experiments*

The PlantDoc dataset and the self-made peanut rust dataset were selected to evaluate the generalization and transfer learning ability of the optimized model. To explore the influence of the image processing method on the test results, the peanut rust dataset was only processed by methods such as shearing and angular rotation (90°, 180°, 270°) that did not affect the image features to obtain the original dataset. Furthermore, the dataset was processed by methods of shearing, angular rotation, and data augmentation (methods described in 2.2) to obtain an equal number of improved datasets. The datasets were input into the network for the transfer learning experiment, and the experiment results are shown in Figures 15 and 16. The classification precision of the improved dataset was 92.57%, which was 7.92% higher than the original dataset, indicating that the image processing method had an impact on the precision of the model. The main reason is that the data enhancement operation can enrich the information of the feature layer and avoid the problem of overfitting the model due to the overlapping of features. The average accuracy of the model on the PlantDoc dataset and the self-made improved dataset reached 90.26% and 92.57%, respectively, indicating that the optimized model proposed in this paper had good generalization ability and could identify different kinds of plant diseases through transfer learning, which provided a reference for expanding the application scope of the model in the future.

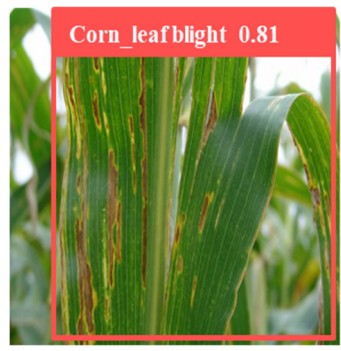
(**a**) Corn_leaf blight (0.81)

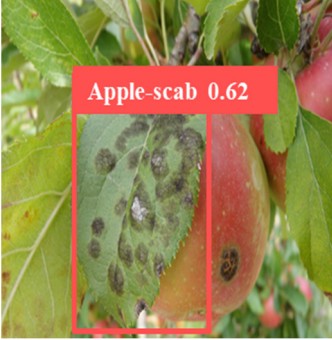
(**b**) Apple-scab (0.62)

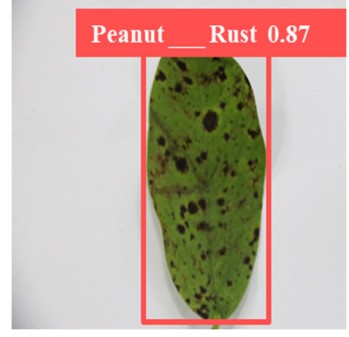
(**c**) Peanut_Rust (0.87)

**Figure 15.** Partial example images of classification results of transfer learning experiments. Representing the disease type (confidence) are Corn_leaf blight (0.81) (**a**); Apple-scab (0.62) (**b**); Peanut_Rust (0.87) (**c**), respectively.

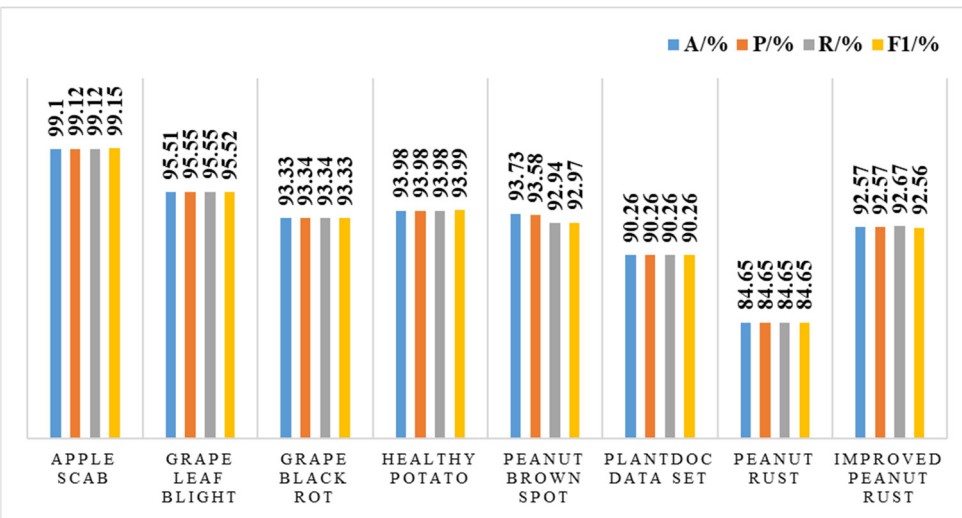

**Figure 16.** Detection results of different datasets (apple scab, grape leaf blight, grape black rot, healthy potato, peanut brown spot, PlantDoc dataset, peanut rust, and improved peanut rust).

Based on the analysis of the above experimental results, considering that manual plant disease monitoring is laborious and error-prone, the use of computer vision and artificial intelligence (AI) for early detection of plant diseases can help reduce the adverse effects of diseases and overcome the shortcomings of continuous manual monitoring. Our findings propose an attention-based deep learning model for the identification of peanut diseases (brown spot and rust). The experimental results show that the performance of the model (93.73%, 92.94%, and 92.97%, respectively, for accuracy, recall, and F1 score) is better than the current mainstream target detection model; the improved method introduced (IASM mechanism, BiFPN structure, Ghostnet and WBF structure) has a relatively good effect; the model has good transfer learning ability (the detection accuracy is 7.92% higher than the original data set). The recognition accuracy is improved when the detection speed is higher than the original model, which finally provides technical support for further research related to plant disease recognition.

### 5. Conclusions

This study proposes a migration learning method for plant disease identification and classification by improving the YOLOv5 model by proposing the improved attention submodule (IASM), introducing Ghostnet structure and the weighted frame (WBF), and applying BiFPN structure and Fast normalized fusion. First, the collected images of peanut brown spots and rust were marked and processed, and a dataset of peanut disease images was created to expand the dataset and reduce the degree of model overfitting. Secondly, according to the structure of YOLOv5 and the characteristics of the dataset, the model structure was optimized and improved. Then, the PlantVillage dataset and the self-made datasets were utilized to test the optimized YOLOv5 model. Finally, the relevant experiments on the model performance were carried out: (1) the performance comparison test between the optimized model and other models (Faster R-CNN, VGG16, SSD, Efficient-Det, YOLOv4, and YOLOv5) was carried out. The operation time and precision were 15 ms-93.73%, which were 93.6–9.5%, 90.5–7.07%, 81.0–8.36%, 53.13–2.88%, 37.5–7.14%, and 11.8–3.98% higher than those of the Faster R-CNN, VGG16, SSD, EfficientDet, YOLOv4, and YOLOv5 model, respectively, indicating that the model proposed in this study achieved a good performance. (2) The ablation test was carried out to verify the effectiveness of the fusion IASM mechanism, the Ghostnet structure, and the BiFPN structure improvement method in this paper. The F1 score of the optimized model reached 92.97%, which was 0.59% and 0.9% higher than the IASM +YOLOv5 model and the BiFPN + YOLOv5 model, respectively, which proved that the two improvement methods used in this study were

effective for model optimization. (3) The PlantDoc dataset and the self-made peanut rust dataset were used to test the generalization ability of the model, and the accuracy reached 90.26% and 92.57%, which showed that the optimized model had better generalization ability and provided a reference for expanding the application scope of the model in the future.

In conclusion, in this paper, the method of crop disease identification and classification was improved and proposed, and the effectiveness of the optimized model in plant disease detection and classification was verified through experiments. Future work will focus on the following:

(1) To expand the application scope of the model, more datasets of plant diseases should be collected. Since there is still room for improvement in the quantity and quality of self-made datasets, the recognition rate of some images is low due to inconspicuous classification features. Further research is needed on image recognition with inconspicuous classification features. Therefore, the self-made dataset will be expanded next. It uses the transfer learning ability of the model to identify and detect more kinds of plant diseases and provides a faster and more efficient plant disease detection scheme for actual needs.

(2) The loss function of the improved recognition method includes the positioning error of the detection frame, and the positioning error in the detection of small targets has a more significant impact on the loss function. Therefore, it is necessary to optimize and improve the model according to the characteristics of small target recognition to improve the accuracy further.

(3) Further studies in other fields, such as soil science and tropical crop productivity, can be used to explore the relationship between the effects of plant diseases and expand the guiding role of model output results on agricultural production.

**Author Contributions:** Conceptualization, H.W. and S.S.; methodology, H.W.; software, H.W.; validation, H.W., X.H. and H.Z.; formal analysis, H.W.; investigation, K.F. and S.S.; resources, K.F. and D.W.; data curation, X.H.; writing—original draft preparation, H.W.; writing—review and editing, H.Z.; visualization, H.W. and H.Z.; supervision, H.W. and D.W.; project administration, K.F.; funding acquisition, S.S. and D.W. All authors have read and agreed to the published version of the manuscript.

**Funding:** This research was funded by the National Modern Agricultural Industry Technology System Post Scientist Project (CARS-13-National Peanut Industry Technology System-Sowing and Field Management Mechanization Post), Shandong Province Major Science and Technology Innovation Project (2021CXGC010813), the high-efficiency ecological agriculture innovation project of Taishan Industrial Leading Talents Project (LJNY202104). Postgraduate Innovation Program of Qingdao Agricultural University (QNYCX21021).

**Institutional Review Board Statement:** Not applicable.

**Informed Consent Statement:** Not applicable.

**Data Availability Statement:** Not applicable.

**Acknowledgments:** We would also like to take this opportunity to thank Shang for providing support with the computing resources.

**Conflicts of Interest:** The authors declare no conflict of interest.

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
