# Peer review of "Plant Disease Detection and Classification Method Based on the Optimized Lightweight YOLOv5 Model"

_agriculture, doi:10.3390/agriculture12070931_

Round 1

Reviewer 1 Report

Reviewer

MDPI – Agriculture

Manuscript Number: ID 1755631

Title: Plant disease detection and classification method based on the optimized lightweight YOLOv5 model

As requested, I have reviewed the above-titled paper for the Agriculture - MDPI Journal. I divided my comments in the sections presented as follows.

Contribution

This paper investigates a method to identify plant disease and classification taking as baseline the YOLOv5 model including the mechanism IASM, the Ghostnet structure and weighted frame (WBF),  BiFPN structure and Fast normalized fusion. The authors claim that their approach is unique and discuss such improvement made with respect to previous literature published.

Images of peanuts were stored and addressed to identify diseases. The authors refer also to the procedure of expanding the dataset in order to reduce the degree of model overfitting.  Accordingly to the structure of the YOLOv5 model and corresponding datasets, the initial proposed structure was optimized. Furthermore, the PlantVillage dataset and the self-made datasets were used to test the optimized YOLOv5 model.

As a final step, experiments on model performance were conducted. In this final step, comparisons of the optimized YOLOv5 model were established with other models, namely, Faster R-CNN, VGG16, SSD, EfficientDet, YOLOv4 and YOLOv5.

The authors report on some statistical metrics that were used to show the better performance of the optimized YOLOv5 in contrast to the other aforementioned models.

In special, the authors state that “the optimized model had better generalization ability and provided a reference for expanding the application scope of the model in the future”. The authors also address two potential future lines of work in the final part of the manuscript, highlighting such contribution of the manuscript.

I found the manuscript has an interesting goal to be pursued and presents the conditions to be published in the MDPI – Remote Sensing Journal. However, the text needs to be revised.

In general, the basic ideas, concepts and assumptions are presented along the manuscript but not necessarily in the best order. The manuscript is not clear enough along the development of some of the steps proposed in the methodological approach and with respect to some of the results presented. There are some missing links and the manuscript does not present a clear work flow, even though the authors describe most of the procedures. The edition of the text should also be fully revised.

Taking that into account, there are further questions and doubts I would like to hear from the authors. The reader needs to have those points better clarified.   Figures and Tables should be improved and possibly complemented to adequately present results. That might also lead to explore or reflect about different scenarios still not well and thoroughly explored by the authors in the proposed paper but that deserves attention.

Please, see further comments in the attached file. 

Author Response

We are honored to receive your review and comments, which are of great help to the improvement of the paper. Based on your suggestions and comments, we have made the following revisions to the paper.

The Reviewer's suggestions for the structure of the paper are helpful for the manuscript to better reflect the workflow. The comments on the proposal of increasing the network improvement method of the paper and the thinking on the application scope of the model can well support the possibility of model transfer learning in this paper, and provide support for expanding the application scope of the model.

Thanks again for your review and comments.

Reviewer 2 Report

This manuscript reports on a study of Plant disease detection and classification method based on the optimized lightweight YOLOv5 model. The study design meets the general standards and from what I can judge the data is being collected and analyzed appropriately. This work is an unpublished manuscript with relevant information in a scientific journal for discussion among scientists working in the field.

However, there are some comments should be considered, in this way, the social and scientific relevance of the manuscript would be improved:

Remove the parentheses from each citation in the text, it should be: [1]

Line 129: remove duplicate word: data

Fig. 10. improve image quality and sharpness

Fig.12. add the identification of each image with a letter. The title of each image in red color is diffuse

Fig. 15. improve image quality and sharpness

Line 623: the references must be adjusted to the format and style of the editorial standards of the journal

Discussion

Line 557: I continue to add a paragraph that summarizes the importance, usefulness and social relevance, contemporary of the study, specifically pointing out the Impact, Benefit and Projection, something like this (for example):

There is scientific evidence about the accuracy of the proposed architecture described in the studies by Li et al. [44], in which YOLO-JD has achieved the best detection accuracy, with an average mAP of 96.63%. A similar finding was reported by Islam et al. [45] which obtained an accuracy of 99.43%, but use transfer learning to train the CNN, proposed by the authors, using VGG16. Also, the research by Lee et al. [46], obtained an accuracy of 99%; in this study, an adjustment was made to the images, removing the background and leaving only the potato leaf in the image for subsequent CNN training; on the other hand, tropical disease detection studies in bananas using the Random Forest algorithm to accurately identify soil properties associated with disease symptoms reported accuracy values ​​of around 85.4% [47, 48, 49]. Therefore, it is possible to effectively use the proposed architecture where the learning transferability of the model can be used to extend the scope in the long term.

Early detection of plant diseases using computer vision and artificial intelligence (AI) like the one applied in this study and in other areas of soil science and tropical crop productivity [50, 51] can help reduce the adverse effects of diseases and also overcome the shortcomings of continuous human monitoring, considering that manual plant disease monitoring is laborious and error-prone. The results of our study propose a deep learning model based on the attention mechanism for the recognition of peanut disease (brown spot and rust). Recognition accuracy is improved at detection speed higher than that of the original model, finally, our results provide technical support for further research related to plant disease recognition.

References

I suggest adding recent references which address the issue in question in Latin American territories. I suggest incorporating the new recommended references in the discussion section, that would improve the scientific quality of the candidate manuscript. Suggested citations are for genuine scientific reasons that emphasize the current topic of study in context:

[44] Li D, Ahmed F, Wu N, Sethi A.I. YOLO-JD: A Deep Learning Network for Jute Diseases and Pests Detection from Images. Plants, 2022, 11, 937. https://doi.org/ 10.3390/plants11070937  

[45] Islam F, Hoq M.N, Rahman CM. Application of transfer learning to detect potato disease from leaf image. IEEE International Conference on Robotics, Automation, Artificial-Intelligence and Internet-of-Things (RAAICON). 2019, p.127-130,2019, https://doi.org/10.1109/RAAICON48939.2019.53

[46] Lee, T.Y., Yu, J.Y., Chang, Y.C., Yang, J.M. HealthDetection for Potato Leaf with Convolutional NeuralNetwork. Indo - Taiwan 2nd International Conference on Computing, Analytics and Networks, Indo-Taiwan ICAN2020 – Proceedings, 2020, p.289–293,. https://doi.org/10.1109/Indo-TaiwanICAN48429.2020.9181312

[47] Rey JC, Olivares B, Lobo D, Navas-Cortés JA, J.A. Gómez, and B.B Landa, Fusarium Wilt of Bananas: A Review of Agro-Environmental Factors in the Venezuelan Production System Affecting Its Development. Agronomy, 2021, 11(5),986, https://doi.org/10.3390/agronomy11050986

[48] Campos O, F. Paredes, J. Rey, D. Lobo, S. Galvis-Causil, The relationship between the normalized difference vegetation index, rainfall, and potential evapotranspiration in a banana plantation of Venezuela. STJSSA, 2021, 18(1), 58-64, http://dx.doi.org/10.20961/stjssa.v18i1.50379

[49] Orlando, O.; Araya-Alman, M.; Acevedo-Opazo, C.; Cañete-Salinas, P.; Rey, J.C.; Lobo, D. & Landa, B. Relationship Between Soil Properties and Banana Productivity in the Two Main Cultivation Areas in Venezuela. J. Soil Sci. Plant Nutr, 2020, 20(3), 2512-2524.  https://doi.org/10.1007/s42729-020-00317-8.

Author Response

We are honored to receive your review and revision comments, which are of great help to the improvement of the paper. Based on your suggestions and comments, we have made the following revisions to the paper.

The Reviewer's suggestions for the structure of the paper are helpful for the manuscript to better reflect the workflow. The comments on the proposal of increasing the network improvement method of the paper and the thinking on the application scope of the model can well support the possibility of model transfer learning in this paper, and provide support for expanding the application scope of the model.

Thanks again for your review and comments.

Round 2

Reviewer 1 Report

Reviewer

MDPI – Agriculture

Manuscript Number: ID 1755631-V2

Title: Plant disease detection and classification method based on the optimized lightweight YOLOv5 model

As requested, I have reviewed the revised version of the above-titled paper for potential publication in the Agriculture - MDPI Journal. I divided my comments in the sections presented as follows.

Contribution

This paper investigates a method to identify plant disease and classification taking as baseline the YOLOv5 model including the mechanism IASM, the Ghostnet structure and weighted frame (WBF),  BiFPN structure and Fast normalized fusion. The authors claim that their approach is unique and discuss such improvement made with respect to previous literature published.

Images of peanuts were stored and addressed to identify diseases. The authors refer also to the procedure of expanding the dataset in order to reduce the degree of model overfitting.  Accordingly to the structure of the YOLOv5 model and corresponding datasets, the initial proposed structure was optimized. Furthermore, the PlantVillage dataset and the self-made datasets were used to test the optimized YOLOv5 model.

As a final step, experiments on model performance were conducted. In this final step, comparisons of the optimized YOLOv5 model were established with other models, namely, Faster R-CNN, VGG16, SSD, EfficientDet, YOLOv4 and YOLOv5.

The authors report on some statistical metrics that were used to show the better performance of the optimized YOLOv5 in contrast to the other aforementioned models.

In special, the authors state that “the optimized model had better generalization ability and provided a reference for expanding the application scope of the model in the future”. The authors also addressed in the first version of the manuscript two potential future lines of work, highlighting such contribution of the manuscript. In the new version, the authors added a third research line to foster new developments in this area.

In addition, the authors presented fair reflections with respect to some comments made regarding the first version of the manuscript and were able to improve the manuscript with some adjustments that have been made in the text, Tables and Figures.

Please, see further comments in the attached file. 

Author Response

We are honored to receive your review and comments again, which have greatly contributed to the improvement of the paper. After previous revisions, the article is more convincing. Based on your suggestions and comments, we have made revisions to the paper.

Relevant grammatical issues have been revised and the full text has been checked and improved. The reviewer's suggestions for the structure of the paper will help the manuscript to be better formatted, consistent with the scientific nature of the article, and meet the format requirements of the journal.

Thanks again for your review and comments.
